# Nuclear SOD1 in Growth Control, Oxidative Stress Response, Amyotrophic Lateral Sclerosis, and Cancer

**DOI:** 10.3390/antiox11020427

**Published:** 2022-02-21

**Authors:** Joyce Xu, Xiaoyang Su, Stephen K. Burley, X. F. Steven Zheng

**Affiliations:** 1Rutgers Cancer Institute of New Jersey, Rutgers, The State University of New Jersey, 195 Little Albany Street, New Brunswick, NJ 08903, USA; jx182@rwjms.rutgers.edu (J.X.); xs137@rwjms.rutgers.edu (X.S.); stephen.burley@rcsb.org (S.K.B.); 2Department of Medicine, Robert Wood Johnson Medical School, Rutgers, The State University of New Jersey, 125 Paterson Street, New Brunswick, NJ 08901, USA; 3RCSB Protein Data Bank, Institute for Quantitative Biomedicine, Rutgers, The State University of New Jersey, 174 Frelinghuysen Road, Piscataway, NJ 08854, USA; 4Department of Chemistry and Chemical Biology, Rutgers, The State University of New Jersey, 174 Frelinghuysen Road, Piscataway, NJ 08854, USA; 5RCSB Protein Data Bank, San Diego Supercomputer Center, University of California, San Diego, 9500 Gilman Drive, La Jolla, CA 92093, USA; 6Department of Pharmacology, Robert Wood Johnson Medical School, Rutgers, The State University of New Jersey, 675 Hoes Lane, Piscataway, NJ 08854, USA

**Keywords:** superoxide dismutase 1 (SOD1), reactive oxidative species (ROS), cell signaling, transcription, ribosome biogenesis, cancer, amyotrophic lateral sclerosis (ALS)

## Abstract

SOD1 is the major superoxide dismutase responsible for catalyzing dismutation of superoxide to hydrogen peroxide and molecular oxygen. It is well known as an essential antioxidant enzyme for maintaining cellular redox homeostasis. SOD1 dysregulation has been associated with many diseases, including amyotrophic lateral sclerosis (ALS), cancer, accelerated aging, and age-related diseases. Recent studies also revealed that SOD1 can serve as a regulatory protein in cell signaling, transcription, and ribosome biogenesis. Notably, SOD1 is localized in the nucleus under both normal and pathological conditions, contributing to oxidative stress response and growth control. Moreover, increasing evidence points to the importance of nuclear SOD1 in the pathogenesis of ALS and cancer.

## 1. Introduction

Superoxide (O_2_^−^) is an anion free radical primarily produced in mitochondria due to leakage during electron transport. As much as one to two percent of oxygen consumed by mammalian cells is estimated to be converted to superoxide [1]. In addition, cytochrome P450 enzyme, NADPH oxidase, and xanthine oxidase are also known to catalyze reactions that produce superoxide. The superoxide anion radical is rapidly converted to hydrogen peroxide (H_2_O_2_), which is comparatively less reactive. Both superoxide and hydrogen peroxide are key reactive oxygen species (ROS) that play important roles in regulatory processes and stress responses [2]. Under normal physiological conditions, they are maintained at low nanomolar range and serve as major signaling agents through modification of protein targets, which regulates growth, metabolic, and stress pathways [2].

Supraphysiological concentration of superoxide and hydrogen peroxide, and their more reactive products such as hydroperoxyl radical (HOO•), hydroxyl radical (HO•), and reactive aldehydes, can oxidize DNA, RNA, lipids, and proteins, producing not only aberrant regulatory responses to alter cellular processes, but also reversible and irreversible damage to cells and tissues [3]. Oxidative stress can lead to abnormal cell and tissue function, and in extreme cases, lead to growth arrest and cell death [3]. Oxidative stress is a major causative mechanism for many common human diseases [3]. For example, oxidative DNA damage contributes to mutagenesis and tumor initiation, development, and progression; oxidative stress causes loss of pancreatic beta-cells and development of type II diabetes mellitus and associated peripheral nerve damages [4]; oxidative stress is also responsible for neuronal cell death, and development and progression of motor neuron [5] and central nervous system (CNS) diseases such as Alzheimer’s and Parkinson’s diseases [6].

Because of its severely deleterious effects, superoxide is rapidly dismutated through enzyme-catalyzed reactions into hydrogen peroxide, that in turn is converted to water and molecular oxygen. Superoxide dismutases (SODs) are enzymes responsible for scavenging superoxide into hydrogen peroxide and molecular oxygen via the following chemical reaction: 2O_2_•^−^ + 2H^+^ → H_2_O_2_ + O_2_ [7]. All cellular organisms express SODs with heavy metals in their catalytic centers. Mammalian cells possess three SOD isoforms: SOD1, a Cu^2+^/Zn^2+^ enzyme; SOD2, a Mn^2+^ enzyme; and SOD3, a distinct Cu^2+^/Zn^2+^ enzyme [7]. The three SODs are distinct in terms of their cellular distributions: SOD1 and SOD2 are both intracellular proteins [7], whereas SOD3 is a secreted protein (also known as extracellular SOD, EcSOD) [8]. SOD1 can be found throughout the cytosol, mitochondrial intermembrane space, and nucleus. In contrast, SOD2 occurs exclusively in the mitochondrial matrix. Since SOD2 and SOD3 are spatially restricted by compartmentalization, not surprisingly, SOD1 has been found to be a primary regulator of redox signaling in diverse cellular processes that occur in the plasma membrane, cytoplasm, and nucleus. Remarkably, SOD1 is prominently localized in the nuclei of different tissues and cell types as well as under disease conditions. Hence, understanding the biological and pathological roles of nuclear SOD1 is of physiological and pathological significance. This review will focus on growth control, oxidative stress response, amyotrophic lateral sclerosis, and cancer in which nuclear SOD1 has been studied. We hope this article will spur interest in studying nuclear SOD1 in other biological systems.

## 2. Role of SOD1 in Normal Physiology and Diseases

SOD1 was discovered more than 50 years ago by biochemical purification of the enzymatic activity catalyzing dismutation of superoxide ions [9]. SOD1 has been associated with many diseases, including premature aging and aging-associated pathologies in mice, progressive motor neuron denervation, muscle wasting, reduced fertility, macular degeneration, and increased incidence of tumors [10]. Germline mutations in SOD1 have been implicated in familial amyotrophic lateral sclerosis (fALS), an adult-onset neurodegenerative disorder that affects motor neurons [11]. Indeed, about 20% all fALS cases carry germline mutations in SOD1 [12]. Many of the fALS-associated mutations result in SOD1 protein aggregation [13,14]. Because *Sod1* knockout in mice fails to recapitulate motor neuron disease phenotypes, fALS mutations in SOD1 are thought to give rise to a gain-of-function causing motor neuron degeneration [13,14].

SOD1 is overexpressed in many human cancers, including non-small-cell lung cancer (NSCLC), breast cancer, and nasopharyngeal carcinoma (NPC) [15,16,17,18]. Moreover, increased expression of SOD1 is correlated with disease progression and poor prognoses in these cancers [19,20]. Knockdown or pharmacological inhibition of SOD1 has been shown to attenuate cancer cell growth in vitro, which show SOD1 is necessary for proliferation of breast cancer, NSCLC, nasopharyngeal carcinoma (NPC), and leukemia cells [7]. Genetic or pharmacological inhibition of SOD1 significantly reduces tumor development in vivo [7]. In inducible erbB2 (MMTV-iErbB2) and Wnt (MMTV-Wnt) transgene breast cancer models in mice, *Sod1* knockout attenuates these oncogene-driven proliferations, but not normal proliferation of the mammary gland associated with pregnancy or other normal proliferative tissues, such as skin and gastro-intestinal tract epithelium [16]. Similarly, in a Kras/TP53 lung tumor-initiation model, tamoxifen-inducible *Sod1* knockout significantly reduced lung tumor burdens due to inhibition of cancer cell proliferation [21]. In contrast, *Sod1* knockout does not have discernible effect on proliferation of normal lung cells or other tissues [21]. These observations document that SOD1 is an oncogene.

Given its role in cancer cell proliferation, SOD1 represents a promising anticancer drug target. Several distinct small molecule inhibitors of SOD1 have been identified that are under preclinical and clinical evaluations, including ATN-224 and LCS-1. ATN-224, a copper chelator, is the most advanced SOD1-targeting agent that has been tested in Phase 1 and Phase 2 clinical trials for patients with relapsed prostate cancer [22,23,24]. Regrettably, none of these trials met their desired endpoints, which may be due to lack of selectivity. More mechanistic studies are required to fully understand the roles of SOD1 in cancer to discover and develop efficacious anti-cancer therapeutics targeting SOD1.

*Sod1* knockout mice exhibit widespread oxidative stress, elevated oxidative DNA damage, and shortened lifespans [25]. These mice develop liver tumors late in life [25], likely as a result of tissue-damage-induced liver inflammation and disease progression. Similarly, in a Kras/TP53 lung tumor initiation model, tamoxifen-inducible *Sod1* knockout increased tumor incidence [21]. These observations suggest that SOD1 plays a role in tumor initiation, which is consistent with SOD1’s principal biochemical function in reducing oxidative stress. Interestingly, SOD1 is phosphorylated by mTOR in response to stimulation, which suppresses SOD1 enzymatic activity [26]. In the tumor microenvironment, cancer cells often are deprived of nutrients due to inadequate blood supply. Under conditions of nutrient starvation, SOD1 becomes dephosphorylated and activated, enabling cancer cells to be more resistant to oxidative stress and enhancing their survival. This mTOR-dependent mechanism maintains redox homeostasis and allows cancer cells to switch between growth and survival mode in response to nutrient availability in the tumor microenvironment [26].

## 3. Localization and Regulation of SOD1 Protein in the Nucleus

Early studies using subcellular fractionation showed that SOD1 is present in the cytosol, nucleus, peroxisomes, and mitochondrial intermembrane space of rat liver [27]. Immunofluorescent analyses revealed that SOD1 is strongly localized in the nuclei of yeast and human fibroblasts [28], HeLa cervical cancer cells [29], human peripheral blood mononuclear cells (PBMCs), human neuroblastoma cells [30], human airway lung cells, and human and mouse NSCLC cells [21]. Immunohistochemistry (IHC) staining showed that nuclear SOD1 protein occurs in various human tissues, including different regions of the brain, intestines, kidney, liver, lung, pancreas, and testis (https://www.proteinatlas.org/ENSG00000142168-SOD1/tissue, accessed on 18 January 2022). Nuclear SOD1 is particularly enriched in the breast, cerebellum, the dermis layer of the skin, and testis.

In yeast and human fibroblasts, SOD1 becomes more enriched in the nucleus in response to oxidative stress elicited by increased H_2_O_2_ levels [28]. In yeast, translocation of SOD1 into the nucleus is regulated by the Mec1/ATM-Dun1/CHK2 kinase pathway [28]. ATM kinase is an oxidative sensor directly activated by H_2_O_2_ [31]. In yeast, upon activation of Mec1/ATM by ROS, Dun1/CHK2, a protein kinase downstream of Mec1/ATM, binds to SOD1 and phosphorylates SOD1 at S60, 99, which promotes SOD1 nuclear localization [28]. In humans, H_2_O_2_ was also shown to elicit SOD1 nuclear accumulation in human fibroblasts and HeLa cells [28,32]. This effect occurs in wild-type, but not *ATM*-deficient fibroblasts [28], suggesting that the ATM pathway is a conserved regulator of SOD1 nuclear localization. In human PBMCs and neuroblastoma cells, activation of ATM/CHK2 and ATR/CHK1 DNA pathways leads to phosphorylation of SOD1 and nuclear accumulation of phosphorylated SOD1 [30]. It should be noted that an ATM/CHK2-dependent phosphorylation site(s) in human SOD1 has yet to be identified.

Protein transport into the nucleus is mediated by short amino acid sequences known as nuclear localization sequences (NLSs), which are recognized by importins [33]. Similarly, protein export is carried out by exportins, through binding to nuclear export sequences (NESs) of cargo proteins [33]. Because SOD1 protein does not contain an obvious classical nuclear localization sequence (NLS), SOD1 nuclear localization could be mediated by an unconventional nuclear localization sequence, or through interaction with another nuclear protein. It was shown that partial unfolding of SOD1 proteins exposes an otherwise normally buried nuclear export sequence (NES), which leads to CRM1 exportin-mediated clearance of misfolded SOD1 from the nucleus [29]. It remains to be determined what mechanism is responsible for export of SOD1 from the nucleus.

## 4. Nuclear SOD1 Acts as a Transcription Factor during the Oxidative Stress Response

In yeast cells, hydrogen peroxide stimulates SOD1 entry into the nucleus and regulates the expression of oxidative-stress-responsive genes. SOD1 target genes include those involved in oxidative stress resistance (e.g., the cytoplasmic and mitochondrial thioredoxin peroxidases Tsa2 and Prx1), DNA damage repair and genomic stability maintenance (e.g., Rad16, a subunit of the nucleotide excision repair factor 4), DNA replication stress (e.g., the large subunit of ribonucleotide-diphosphate reductase Rnr3), and iron/copper homeostasis (e.g., the iron/copper reductases Fre1/3/8). Consistent with a role of nuclear SOD1 in DNA damage repair and maintenance of genomic stability, nuclear SOD1, not cytoplasmic SOD1, was shown to provide protection against DNA damage in yeast and chicken cells [28,34].

Mechanistically, yeast SOD1 was shown to bind to promoters of its target genes, indicating that SOD1 acts as a transcription factor (Figure 1). In a large-scale profiling for human DNA-binding proteins using a protein microarray, over 300 unconventional DNA-binding proteins were identified, including SOD1 [35]. In the same study, the consensus DNA-binding motif for SOD1 was predicted to be 5′-GACGACGAA-3′. Moreover, SOD1 binds this motif in an electrophoresis mobility shift assay (EMSA). In another study, chromatin immunoprecipitation sequencing (ChIP-seq) analysis revealed that there is a ROS-dependent redistribution of SOD1 binding sites in HeLa cells [32]. Several SOD1 consensus DNA-binding motifs were derived in the same study. Curiously, these SOD1-binding motifs do not closely match the consensus sequence identified earlier in the protein microarray study. One possible explanation for the difference is that there is a cell type-specificity toward SOD1’s transcriptional targets. A DNase I footprinting assay showed that SOD1 is associated with DNA through direct contact with the base sequence 5′-GGA-3′ or 5′-GAA-3′ [32]. Based on solution small-angle X-ray scattering (SAXS) structure and computational modeling of the three-dimensional structure of the SOD1:DNA complex, SOD1 is thought to use a surface α-helix to bind in the major groove of DNA [32].

## 5. Roles of Nuclear SOD1 in Cancer

SOD1 is overexpressed in many human cancers, including non-small-cell lung cancer (NSCLC), nasopharyngeal carcinoma (NPC), and breast cancer [15,16,17,18]. Moreover, increased expression of SOD1 is correlated with disease progression and poor prognoses in these cancers [19,20]. IHC staining showed that SOD1 protein is prominently expressed in the nuclei of all 20 types of human tumors curated in the Human Protein Atlas database (https://www.proteinatlas.org/ENSG00000142168-SOD1/pathology, accessed on 18 January 2022). Consistently, SOD1 is found enriched in the nuclei of breast and lung cancer cells [16,21]. Engineered wild-type and the nuclear form of SOD1, but not the cytoplasmic form of SOD1, suppress the growth defect of *Sod1* knockout mouse NSCLC cells [21], demonstrating that nuclear SOD1 is important for NSCLC cell growth. Superoxide is generated primarily in the mitochondria and membrane-bound NADPH oxidases. However, superoxide radicals are impermeable through the lipid bilayer of the nuclear envelope. Hence, SOD1’s unclear function in cancer cells seem to be unlikely to be directly related to its canonical role as an antioxidant enzyme. Consistently, a comprehensive panel of commonly used cell-permeable antioxidants fail to suppress the growth defect of *Sod1* knockout mouse NSCLC cells, including NAC (N-acetyl-l-cysteine), MnTBAP and Tempol (superoxide dismutase mimetics), GSH-MEE (a membrane permeable GSH derivative), and Trolox (water-soluble analogue of vitamin E) [21]. These observations suggest that nuclear SOD1 carries out a non-canonical function(s) essential for cancer growth and proliferation.

Ribosome biogenesis occurs primarily in the nucleolus [36]. It involves transcription and processing of pre-ribosomal RNAs (pre-rRNAs) into mature rRNAs, and assembly of mature rRNAs with ribosome proteins into 40S and 60S ribosome subunits. During oncogenesis, ribosome biogenesis is strongly up-regulated by various growth-signaling pathways, including mTOR [37,38,39]. The PES1-BOP1-WDR12 (PeBoW) complex is required for 41S pre-rRNA processing necessary for production of the 60S ribosome subunits [40,41]; its overexpression promotes proliferation of cancer cells [40]. SOD1 is partially localized in the nucleolus, where it interacts with the PeBoW complex and maintains the stability of the PeBoW complex [21]. In the absence of SOD1, 41S pre-rRNA processing and 60S ribosome maturation are blocked, causing cancer cell growth arrest. Enlarged nucleoli and nucleolar hypertrophy are characteristic of ribosome biogenesis [42]. The later phenomenon is associated with poor prognosis, and used as a prognostic marker in clinical pathology [42]. Consistent with an essential role of in hyperactive ribosome biogenesis, *Sod1* knockout blunts nucleolar hypertrophy in Kras/TP53-driven NSCLC tumors in mice [21]. These observations demonstrate that regulation of ribosome biogenesis is a mechanism by which nuclear SOD1 promotes cellular growth in cancer.

SOD1 as a transcription factor is also thought to contribute to cancer development and maintenance. Due to aberrant metabolism, cancer cells contain much higher ROS content than that found in normal cells. Although moderate levels of ROS stimulate cancer cell proliferation, excess ROS damages macromolecules, such as DNA and lipids. As described above, SOD1 promotes expression of many genes involved in oxidative stress response. Ribonucleotide reductase (RNR), for example, is a SOD1 transcriptional target [28,32]. This enzyme is essential for synthesis of deoxyribonucleosides (dNTPs) necessary for DNA repair and DNA replication [43]. In yeast, SOD1 is necessary for RNR activity and SOD1 deficiency results in sensitivity to DNA replication stress and DNA damage [44,45]. In HeLa cells, SOD1 has been shown to bind to the promoters of oncogenes and tumor suppressor genes [32]. Notably, SOD1 downregulates the tumor suppressor gene PTK6 and upregulates oncogenes, such as FGFR4, TUSC2, and CRTC3 [32]. These studies show that multiple transcriptional functions of nuclear SOD1 are involved in cancer cell growth and survival (Figure 2).

## 6. Role of Nuclear SOD1 in ALS

ALS is characterized by accumulation of SOD1 protein aggregates that are thought to be toxic to motor neurons [46,47]. Despite considerable efforts over the past two decades, precisely how SOD1 aggregates lead to pathogenesis of the ALS remain not fully understood. Several studies have examined the effect of protein aggregation on SOD1 nuclear localization and functions. Elevated nuclear SOD1 aggregate levels were observed in motor neurons of sporadic amyotrophic lateral sclerosis (sALS) patients, when compared with healthy individuals [48]. Interestingly, misfolding of SOD1 proteins is found to expose an internal NES-like sequence, which is recognized by the exportin CRM1, leading to conveyance of misfolded SOD1 proteins to the cytoplasm [29]. Mutation of the NES-like sequence causes accumulation of mutant SOD1 proteins in the nucleus, and motor neuron cytotoxicity. Moreover, SOD1 with an impaired NES-like sequence impairs locomotion, egg-laying, and survival of Caenorhabditis elegans, phenotypes similar to ALS patients [29]. This study provided direct evidence for neurotoxicity of SOD1 aggregates when they are present in the nucleus.

In PBMCs from sALS patients, SOD1 aggregates prevent hydrogen peroxide-induced ATM/CHK2-dependent translocation of SOD1 into the nucleus, which is correlated with elevated oxidative DNA damage [30]. One possible causative mechanism is that SOD1 aggregates interfere with the function of SOD1 as a transcription factor, preventing up-regulation of oxidative stress response and DNA repair genes (Figure 3). In HeLa cells, SOD1 was shown to bind to the promoters and regulate expression of UNC13A, PTK6, NRG1, and INA genes, which have been implicated in the development and progression of ALS [32] and may contribute to the cytotoxic effect of SOD1 protein aggregation. In a transgenic mouse model expressing human SOD1 (hSOD1) fALS mutant variants (G93A and G37R) that displays fALS clinical and pathological features, prominent nuclear accumulation of both wild-type and mutant hSOD1 was observed in the nuclei of spinal cord cells, particularly motor neurons [49]. However, the survival motor neuron protein (SMN) complex is disrupted in motor neuron nuclei in mutant hSOD1 transgenic mice, but not age-matched wild type hSOD1 transgenic mice. Hence, disruption of the nuclear SMN complex by mutant SOD1 represents another potential mechanism causing ALS.

## 7. Conclusions and Future Perspectives

SOD1 has been known to be localized in the nucleus for over two decades. Only recently did its nuclear functions come to light. Nuclear SOD1 engages in non-canonical biochemical functions, such as regulation of transcription and ribosome biogenesis, both processes important for oxidative defense and growth. Importantly, nuclear SOD1 functions provide new insights into pathological mechanisms of SOD1 in cancer and ALS. Moving forward, it will be important to further understand the molecular details of these new functions in more physiologically relevant in vivo models, which will be necessary for improving prevention and treatment of these diseases. Given the difficulty to treat KRAS-driven NSCLC, targeting nuclear SOD1 should open a new avenue for discovery and development of new targeted therapeutic agents for management of this lethal malignancy.

## Figures and Tables

**Figure 1 antioxidants-11-00427-f001:**
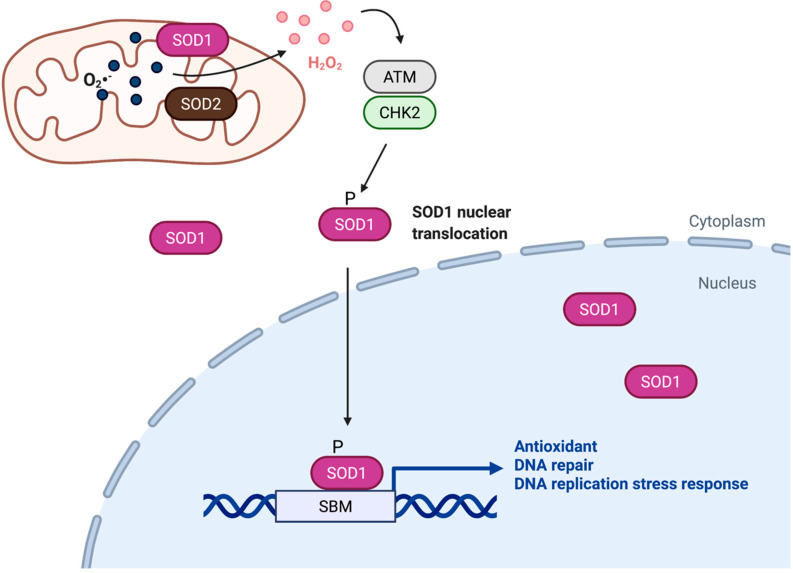
SOD1 is a transcription factor that regulates of redox homeostasis. Upon activation by H_2_O_2,_ ATM-CHK2-dependent phosphorylation promotes accumulation of SOD1 in the nucleus, where SOD1 binds to promoters and regulates expression of genes involved in oxidative stress responses. SBM, SOD1 binding motif.

**Figure 2 antioxidants-11-00427-f002:**
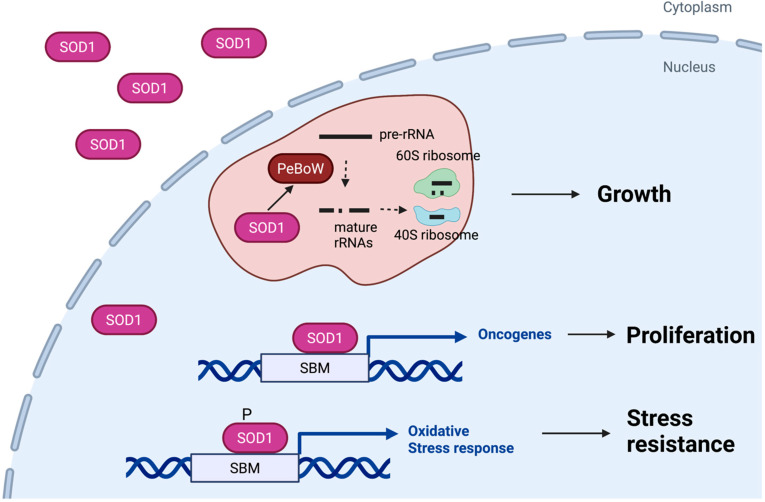
Roles of nuclear SOD1 in cancer. Nuclear SOD1 contributes to growth of cancer cells by at least three distinct mechanisms. First, SOD1 acts as a transcription factor that promotes expression of genes that mitigate oxidative stress in cancer cells. Second, SOD1 binds to gene promoters and regulates expression of multiple oncogenes and tumor suppressor genes. Third, SOD1 is partially localized in the nucleolus and regulates assembly of the PeBoW complex, which stimulates pre-rRNA processing and ribosome biogenesis. SBM, SOD1 binding motif.

**Figure 3 antioxidants-11-00427-f003:**
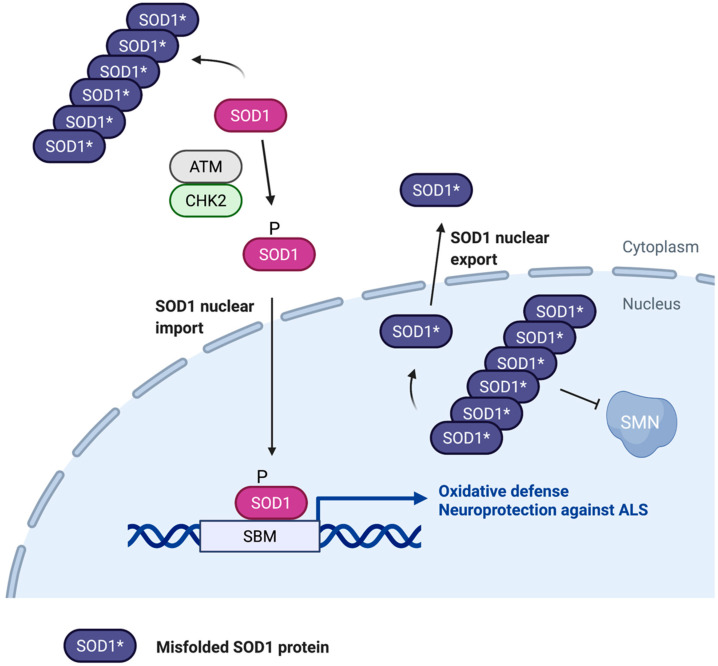
Role of nuclear SOD1 in ALS. Misfolding and aggregation of SOD1 may cause motor neuron cytotoxicity by interfering with nuclear SOD1 localization and function. SOD1 aggregates in the cytoplasm appear to block ATM/CHK2-stimulated nuclear localization and transcriptional activity of SOD1, which is required for oxidative stress resistance and neuroprotection. SOD1 aggregates also occur in the nucleus where they disrupt the SMN complex, contributing to ALS disease progression. CRM1-mediated export of unfolded SOD1 from the nucleus may help alleviate the neurotoxic effects of SOD1 aggregates. SBM, SOD1 binding motif.

## Data Availability

Data is contained within the article.

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
