# Peer review of "Nuclear SOD1 in Growth Control, Oxidative Stress Response, Amyotrophic Lateral Sclerosis, and Cancer"

_antioxidants, 2022, doi:10.3390/antiox11020427_

Round 1

Reviewer 1 Report

This is an excellent review focusing on the novel and very intriguing role of SOD1 as a putative transcription factor and its role in cancer and neurodegenerative disease.  While so many reviews discuss the canonical role of SOD in oxidative stress protection, this review is unique in that it focuses on non-canonical roles of SOD1 and I anticipate will be heavily cited.  I only have very minor suggestions for improvement.

1) in all three figures, the authors show SOD1 binding to ARE.  But ARE is not defined in the main text.  Its a bit confusing because ARE historically is  the antioxidant response element that NRF2 binds.  Some readers might think SOD1 is binding to the same element as NRF2 binds.  The  authors should consider changing the name of the element or at least better defining it so readers do not confuse this with the NRF2 site. 

2.  line 43, can you add hydroxyl radical.

3. line 53, the reference (5) cited connecting ROS to ALS is very old and the thinking in the field has dramatically changed since this review. 

4. line 196 states Superoxide are primarily generated in the cytosol.  This is not necessarily correct.  Most of it is generated in the mitochondria or at the cell surface or other membranes where NADPH oxidases exist. 

Author Response

  1. We have changed ‘ARE’ to ‘SBM’ (SOD1 binding motif) in all three figures and defined ‘SBM’ in figure legends. SBM was previously reported in Nucleic Acid Research (doi.org/10.1093/nar/gkz256).

  1. Hydroxyl radical has been added to line 43.

  1. We would like to update to a more recent review (DOI: 10.1155/2020/5021694). However, we are unable to the Endnote reformat the citation into the publication-formatted manuscript. We hope the journal will help to incorporate this citation.

  1. We have incorporated this suggestion into line 196.

Reviewer 2 Report

The paper focuses on nuclear SOD 1. Interesting subject. Not a lot of papers in this. Definitely of interest for the readers. Written rather straightforward. Lack of literature can partially explain why. Abstract needs reformulation for sure. Why the authors focused on these “ Growth Control, Oxidative Stress Response, Amyotrophic Lateral Sclerosis, and Cancer” and not that other aspects mentioned there. This is actually the case of the entire manuscript which has the info there, but needs to convince more on the aspects they focused, when they wrote it. Anyway, this looks of interest for the Editorial Board (so I will suggest a minor revision), but with a more focused approached on ALS, cancer , (general?) oxidative stress and so on.

Author Response

The focus of this review is on nuclear SOD1 functions in growth control, oxidative stress response, amyotrophic lateral sclerosis and cancer. This is because current literature on nuclear SOD1 is limited to these areas. We elect to not include a comprehensive review of canonical SOD1 functions, in order to avoid unnecessary repeat of what can be readily found in many existing reviews, some of which are cited in our manuscript.